# Reducing Symmetry Mismatch Caused by Freely Placed Cameras in Robotic Learning

## Abstract

Equivariant policy learning has been shown to solve robotic manipulation tasks with minimal training or demonstration data. However, the effectiveness of equivariance depends on whether transformations of the scene align with simple transformations of the input data. This is true when the camera is in a top-down view, but in the common case where a camera views the robot workspace from the side, there is a symmetry mismatch, reducing model performance. We show that equivariant methods perform better when camera images are transformed to appear as top-down images. Our approach is simple to implement, works for RGB and RGBD images, and reliably improves performance across different view angles and learning algorithms.

## 1 Introduction

Policy learning methods in robotics typically rely on expert demonstrations or environment interactions, which are costly and time-consuming to provide. A growing body of work has shown that equivariant networks significantly reduce the amount of data needed to solve manipulation tasks by building symmetry into the model as an inductive bias (Wang et al., 2022c; Simeonov et al., 2022). Since robotics tasks often require generalizing across spatial transformations of the scene or objects, networks that are equivariant to rotations and translations of the input can learn robust policies with significantly less training data than would otherwise be needed.

Most existing work on equivariant robot learning assumes that the symmetry transformations are easy to compute over the input space. This is usually required due to how we create equivariant networks; the action of the symmetry group is used to generate a weight-sharing scheme which constrains the networks layers. However, this can be a problem in robotics when the viewpoint of a physical camera does not align well with the axis about which a rotation symmetry exists in the environment. In particular, an important and standard setting in robotic manipulation involves a camera which views a scene from the side (a "sideview") (Luo et al., 2024; Mandlekar et al., 2023; Brohan et al., 2023), not from the top. This presents a challenge because these robotics domains are generally symmetric about the vertical gravity axis and this axis does not align well with the camera viewpoint (e.g. see Figure 1). The task symmetry is given by rotating and translating objects in the scene, but computing the new sideview image is not a simple image transform due to occlusion and perspective.

Wang et al. (2022b) showed that equivariant networks can still be effective when there is some mismatch between the symmetry group used to constrain the model and the physically accurate task symmetry. Specifically, they found that using image rotations on sideview images to capture $O(2)$ actions on the scene is better than not using equivariance. Nevertheless, there is a noticeable performance gap when compared to the top-down image setting.

In this work, we investigate how to reduce this gap. We find that simple, well-known computer vision techniques can be employed as a preprocessing step to improve performance for $O(2)$ equivariant robot learning. In the case of sideview RGBD images, the depth channel allows us to interpret the image as a point cloud and reproject the image from a top-down view, filling in missing data with interpolation. In the case of RGB images, a homographic perspective transform can approximate a top-down view. These steps can be added to existing equivariant methods and do not require access to privileged information or additional training data. We evaluate the approach on a set of simulated robotic manipulation tasks where the camera views the scene at an angle.

Figure 1: When a camera is freely-placed in a robot environment, there is mismatch between the symmetric transformations of the robot workspace and of the image. This mismatch reduces performance of equivariant methods. The ideal input is a top-down image (blue), but this is not practical since the arm is in the way. We propose using a reprojected image (orange), which converts the sideview image (green) to a top-down view.

The contributions for this work are:

- Two simple image preprocessing steps that reduce the mismatch between image transformations and the true $O(2)$ symmetry in robotic manipulation environments,

- Empirical analysis showing a consistent performance boost for $O(2)$ equivariant robot learning across many settings including reinforcement learning, imitation learning, multiple view angles, and both RGBD and RGB images.

## 2 RELATED WORKS

**Equivariant Robot Learning**    Robot learning is an ideal setting for equivariant networks because data is limited and tasks are almost always equivariant to spatial transformations of the scene. Many works have used $SO(2)$- or $SE(2)$-equivariant convolutional networks with top-down camera images to learn robotic grasping (Zhu et al., 2022), pick and place (Huang et al., 2022), and closed-loop manipulation skills (Wang et al., 2022c; Jia et al., 2023). Other works explored $SO(3)$- or $SE(3)$-equivariant networks that operate on point clouds or voxel grids to learn pick-and-place (Pan et al., 2023; Simeonov et al., 2022; Ryu et al., 2022; 2023; Huang et al., 2024a), or general manipulation skills (Brehmer et al., 2024; Yang et al., 2023). However, the prior works typically require a carefully designed input (i.e., top-down images) to align the input symmetry with the task symmetry. Unlike these works, our effort explores a more general setting where the input symmetry does not align with the task symmetry.

**Learning Latent or Approximate Symmetry**    For some learning problems, there could be a mismatch between the symmetry in the ground truth function and the symmetry in the equivariant network because the symmetry cannot be easily described in the input space or the ground truth function is only partially symmetric. Falorsi et al. (2018) and Park et al. (2022) showed that symmetric neural representations can be extracted using traditional networks with a self-supervised loss. These symmetric representations can be further processed with equivariant layers leading to improved generalization (Esteves et al., 2019; Klee et al., 2023). Another solution to combat this problem is to use approximate or relaxed equivariant neural networks (Wang et al., 2022e; 2024b; Huang et al., 2024b) to relax the equivariant constraint in the network to better match the symmetry in the ground truth function. Alternatively, Wang et al. (2022b) showed that even with the symmetry match, a fully equivariant model that enforces symmetry to out-of-distribution data can still outperform non-equivariant baselines, as long as the symmetry in the model does not conflict with the ground truth function (Wang et al., 2024a). A similar finding was shown in De Silva et al. (2023) where training with out-of-distribution data could aid learning. Although the solution of Wang et al. (2022b) is simple and effective, there remains a significant performance gap compared to not having the symmetry mismatch. Our work provides a simple means to close this gap.

**Image Reprojection in Deep Learning**  Image reprojection, or transforming an image between projected coordinate spaces or camera frames, is a well-known and common technique in computer vision, graphics and robotics. In the context of deep learning, perspective transforms have been used as data augmentation to improve robustness for pose estimation (Mohlin et al., 2020). Jaderberg et al. (2015); Gao et al. (2020) proposed adding affine and perspective transforms with trainable parameters to convolutional networks. In robot learning, some works (Goyal et al., 2023; Ten Pas et al., 2017) have projected images from 3D data like voxels or point clouds as input which enabled much faster training and inference. Lin et al. (2023) trained a Neural Radiance Field representation of a scene then rendered novel view images. A CNN was trained to predict $SO(2)$ manipulation actions in the respective image planes. Our work also uses reprojection to align the action space with the image plane. However, our work does not need to be trained so it is well-suited for settings with limited data.

## 3 BACKGROUND

### 3.1 EQUIVARIANT NETWORKS

An equivariant function preserves symmetry in the input and output spaces. Given a symmetry group $G$ that acts on domain $x \in \mathcal{X}$ and codomain $y \in \mathcal{Y}$ with group representations $\rho_x$ and $\rho_y$, respectively, a function $f$ is equivariant to $G$ if

$$f(\rho_x(g)x) = \rho_y(g)f(x) \tag{1}$$

for all $g \in G$. In other words, applying a transformation to the input produces a transformed output. Notice that the representations $\rho_x$ and $\rho_y$ are assumed to be known in advance in order to design networks that satisfy the equivariant constraint. For example, equivariant convolutional networks are implemented by constructing kernels $K$ that satisfy the equivariant kernel constraint $K(gv) = \rho_{\text{out}}(g)K(v)\rho_{\text{in}}(g)^{-1}$ where $\rho_{\text{in}}$ and $\rho_{\text{out}}$ are the representations for the input and output of the layer (Cohen et al., 2019).

Equivariant networks are composed of equivariant layers and equivariant activation functions. To properly leverage symmetry this way, it is common to assume that the input is structured such that group representations $\rho_x(g)$ can be easily defined and computed. For example, when the input is an image and $G = SO(2)$, a top-down image is typically used where the group representation on the input is defined as image rotation.

### 3.2 EQUIVARIANT POLICY LEARNING

Policy learning is a framework for solving decision problems. The objective is to train a policy network $\pi : S \to A$ that maps from states to actions in order to achieve some desired outcome or behavior. In reinforcement learning, the policy is optimized to select actions that maximize the value of the state action pair, where the value function $Q : S \times A \to \mathbb{R}$ is represented with a separate critic network. In imitation learning, the policy is optimized to match the distribution arising from a dataset of expert demonstrations.

Wang et al. (2022c) showed that many relevant robotics tasks can be expressed as group-invariant MDPs. In these cases, the optimal policy network is equivariant: $\rho_a(g)\pi(s) = \pi(\rho_s(g)s)$ for all $g \in G, s \in S$. Note that the representations $\rho_s$ and $\rho_a$ may flexibly define how observations and action variables should change or remain invariant. A straightforward example is robotic grasping: if the scene is transformed then the optimal grasp pose is transformed in the same way, but the optimal gripper opening remains fixed. The reward function is invariant which means the optimal critic network is also invariant: $Q(s, a) = Q(\rho_s(g)s, \rho_a(g)a)$ for all $g \in G, s \in S$.

### 3.3 EXTRINSIC SYMMETRY

Although prior works (Wang et al., 2022a;d; Jia et al., 2023; Nguyen et al., 2023; Liu et al., 2023) demonstrated promising results with equivariant policy learning in various problem settings, they all require an aligned, top-down image observation, which limits the generalizability of the method. Recently, Wang et al. (2022b) showed that equivariant policy learning still outperforms non-equivariant methods with an arbitrary camera view (this is called extrinsic equivariance because the symmetry

will transform the non-top-down image out-of-distribution), but there is a significant performance gap comparing extrinsic equivariant methods with the ideal top-down setting. In this work, we investigate the performance gap and propose a simple yet efficient method that improves equivariant policy learning with an arbitrary camera angle.

# 4 METHOD

In robotic manipulation tasks, we want the equivariant network to generalize across transformations of the physical scene. In the case of a sideview camera image, there is a mismatch between the transformations in image space and in world space. In this section, we describe two ways to preprocess sideview images, such that the resulting image features transform similarly to the underlying physical scene. Both preprocessing approaches are simple to implement. They do not require any pre-training or modification to the robot or sensor setups, and only require knowledge of the camera intrinsics and extrinsics.

## 4.1 REPROJECTION OF RGBD IMAGES

We know that the ideal case for an equivariant network is an image captured by a top-down camera. When the sideview camera captures RGBD images, we can generate a colored point cloud and render a new image using an imaginary top-down camera (see Figure 2).

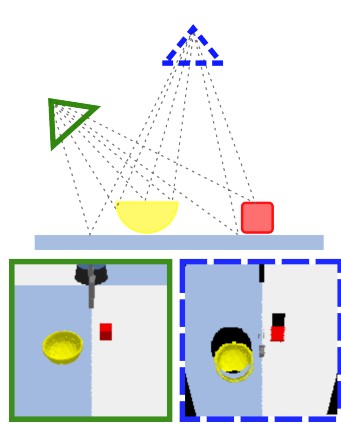

The process for reprojecting an RGBD image onto another image plane is well-known, and implemented in several libraries (Bradski, 2000; Zhou et al., 2018). We describe the process in some detail here. Given camera principal point $(c_x, c_y)$ and focal lengths $f_x, f_y$, we first generate a point cloud in the sideview camera frame from the depth image, $D(u,v)\colon \mathbb{Z}^2 \to \mathbb{R}$, by computing 3D coordinates for each pixel location $(u,v)$

$$x = D(u,v) \cdot (u - c_x)/f_x \tag{2}$$
$$y = D(u,v) \cdot (v - c_y)/f_y \tag{3}$$
$$z = D(u,v) \tag{4}$$

Figure 2: Illustration of RGBD reprojection from sideview (green) to top-down (blue) camera view.

Next, we rotate and translate the point cloud into the frame of a virtual top-down camera, call these new points $(x', y', z')$. Finally, we render the transformed points with an orthographic camera centered above the workspace, as shown here:

$$u' = h \cdot (x'/T + 0.5) \tag{5}$$
$$v' = w \cdot (y'/T + 0.5) \tag{6}$$

where $T$ is the size of the rendered workspace, and $h, w$ are the image dimensions.

This step produces coordinates on the continuous 2D image plane, e.g. $(u', v') \in \mathbb{R}^2$. To generate an image array, we bin all values corresponding to a discrete pixel and select the one with the lowest value of $z'$. We compute RGB color values for the new image by keeping track of which pixel in the original image produced the depth value in the new image. In practice, we discard all points in the point cloud above the gripper height before rendering, to prevent the robot arm from occluding the scene in the reprojected image.

**Occluded Regions** The RGBD image represents a partial point cloud of the scene so the reprojected top-down image will contain empty regions due to occlusion. We assume the original image contains all necessary information to solve the problem so the occlusions should not prevent learning. The occlusions create extrinsic symmetry (Wang et al., 2024a). That is, the occlusions are always located above an object in the image but the equivariant networks expects them to be distributed symmetrically around the object. As extrinsic symmetry can harm performance, we infill the occluded regions with nearest neighbor interpolation (see Appendix A.1). This reduces extrinsic symmetry, but may not be a general solution when the scene is highly cluttered. In robotic manipulation environments, placing the camera higher up (e.g. closer to top-down) reduces occlusion

from objects, but increases occlusion from the gripper and robot arm. We explore this empirically in Section 5.6 and Appendix A.5.

## 4.2 PERSPECTIVE TRANSFORM OF RGB IMAGES

RGB images are a popular input modality for robotic manipulation due to their low cost (Zhao et al., 2023) and ability to capture information at small distances (Luo et al., 2024). An RGB image does not contain spatial information about the scene so we cannot perform the reprojection step from above. Instead, we propose applying a perspective transformation to align the ground plane of the world with the image plane (this technique is sometimes referred to as perspective removal or correction). A perspective transformation is a linear mapping of coordinates between two 2D planes. This linear mapping is described using a homography matrix.

In this setting, we want a mapping that sends the four corners of the robot workspace (3D positions on the ground plane) to the four corners of the image plane (see Figure 3). The first step is to calculate the pixel coordinates of the robot workspace in the image. We transform the 3D position of the workspace corners into the camera frame using the rotation and translation of the camera in the world frame, $R \in \mathrm{SO}(3)$ and $t \in \mathbb{R}^3$, respectively. Then, we project onto the image plane using the camera intrinsic matrix $K$. Given a point $x \in \mathbb{R}^3$, we calculate the pixel location as a homogeneous coordinate $y = K(Rx + t)$.

The next step is to compute the homography matrix given the four workspace corner pixels and the image corner pixels. (We use the OpenCV implementation (Bradski, 2000)). In the final step, we generate a new image by transforming all pixel locations with the homography matrix and interpolate the values onto an image grid.

**Out-of-plane Distortion** A perspective transformation maps features from one 2D plane to another. If the scene captured by the camera only included the ground plane, the transformed image would perfectly represent the scene from a top-down view. In practice, robotic manipulation environments contain 3D objects on top of the ground plane. The appearance of these objects is distorted in the perspective transformation, with more distortion the higher the object is above the plane. The distortions make it harder to learn an policy, so we propose two modifications to the input to enable the policy to resolve the location and appearance of the gripper and objects. We describe the modifications below.

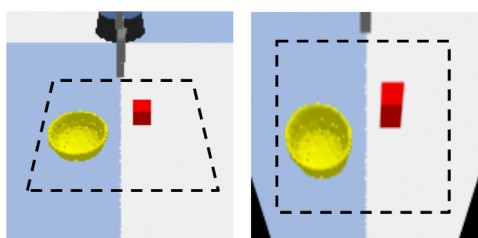

Figure 3: Illustration of perspective transform, with workspace marked as dashed line. Objects above the plane are visually distorted by the transform.

The robotic gripper is substantially distorted and occasionally even shifted outside the frame of view (see Figure 3 right). To re-inject information about the gripper, we concatenate a 'gripper image' along the channel dimension the transformation. The gripper image illustrates where the gripper fingers would lie in the new image plane; the pixel value at the gripper finger locations conveys the gripper height (see Appendix A.1 for a visualization).

The distortion of objects presents a unique challenge to a rotationally equivariant network. Such a network cannot form filters that are specific to an absolute orientation. Thus, an equivariant network will not be able to easily resolve the distortions of objects which always occur toward the top of the original image. To overcome this we encode the camera viewpoint as a 2D vector pointing from the workspace center toward the camera and append this information as an additional two channels. In practice, the two channels are encoded as a $pho_1$ feature, which rotates in 2D as the image is rotated.

## 5 EXPERIMENTS

The image preprocessing steps described in Section 4 reduce the mismatch between how the $O(2)$ group acts on the image and on the environment. However, they also introduce some artifacts or distortions into the image which may harm learning. In this section, we empirically evaluate the benefits of the preprocessing steps for equivariant robot learning from sideview images.

## 5.1 Manipulation Environments

We evaluate our method on six robotic manipulation environments from BulletArm (Wang et al., 2022a), which uses the PyBullet simulator (Coumans & Bai, 2016–2021). The environments are illustrated in Figure 4 and cover various manipulation skills such as picking, pushing and pulling. The state of the environment is an RGB or RGBD image from a perspective camera as well as the gripper position and aperture. The action space is five-dimensional, relative gripper motion in $x, y, z, \theta$, where $\theta$ is rotation about the z-axis, and target gripper aperture $\lambda$. The action space is decomposed into equivariant components $(x, y)$ and invariant components $(z, \theta, \lambda)$. The reward is sparse: +1 if the task is completed and 0 otherwise. All environments are $O(2)$-symmetric; if scene is rotated or reflected in the XY-plane, then the optimal action should be rotated or reflected accordingly. Additionally, the objects in the scene are randomly placed and oriented in the workspace upon reset, so the optimal policy must generalize to unseen spatial configurations.

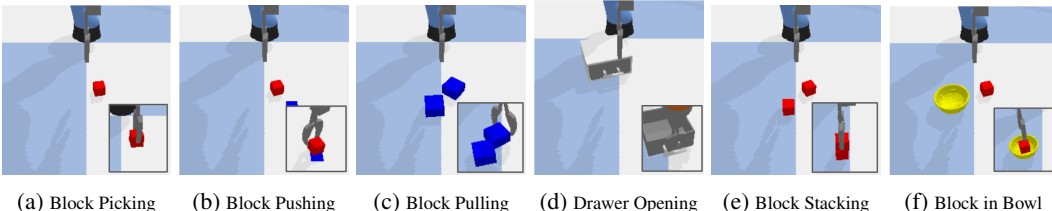

(a) Block Picking    (b) Block Pushing    (c) Block Pulling    (d) Drawer Opening    (e) Block Stacking    (f) Block in Bowl

Figure 4: BulletArm environments. The overlaid images illustrate when the objective of each task.

## 5.2 Policy Learning Settings

We perform experiments in two policy learning settings, reinforcement learning and imitation learning. For reinforcement learning, we use Soft Actor Critic from Demonstrations (SACfD) (Wang et al., 2022c), which prepopulates the replay buffer with some expert demonstrations. SACfD minimizes the original SAC loss terms (Haarnoja et al., 2018) and an L2 loss between the policy prediction and the demonstration data, which speeds up the learning. For imitation learning, we use behavior cloning which minimizes the L2 norm between the policy predictions and the demonstrated actions. In all experiments, demonstration data is generated using the expert planner provided by BulletArm. Additional details on the training process can be found in Appendix A.3.

## 5.3 Baselines

In our experiments, we want to understand whether the proposed preprocessing steps help the equivariant networks learn. In the results section, we refer to networks that use reprojection as *Reproj. Equi* and perspective transform as *Persp. Equi*. We directly compare to using the un-aligned, side-view image with the equivariant network (*Sideview Equi*) as used in Wang et al. (2022b). By default, the sideview image is generated with a camera angled at 45 degrees above the horizon. Additionally, we baseline against an oracle observer *Oracle Equi*, which sees the environment from a top-down view. These images have perfect symmetry match, so we expect the equivariant networks to perform best here. We refer to this baseline as an oracle, because it is almost impossible to achieve such images in the real-world (multiple RGBD camera views must be fused, robotic mesh removed and the image re-rendered). In addition, for the RGBD observation case, we include comparisons to equivariant networks that take point clouds and voxel grids as input (which are transformed to world space so there is perfect symmetry match). Finally, we compare to a non-equivariant baseline (*Sideview NonEqui*) as a reference point, which takes sideview images as input.

All methods use networks with a similar number of trainable parameters and follow the same training scheme (data augmentation, learning rate, etc.). See Appendix A.4 for more details on the network architectures.

## 5.4 Evaluation with RGBD Images

The learning curves for SACfD with RGBD images are shown in Figure 5. We find that the proposed re-rendering step boosts performance across all tasks. The performance boost is greatest for block

stacking, which is the hardest of the six tasks. Across all environments, the equivariant method with the oracle observation learns an effective policy fastest. The slight performance gap between *Oracle* and *Reproj. Equi* indicates that the occluded regions slow learning. Switching from nearest neighbor interpolation to a pre-trained inpainting model may narrow this gap. The equivariant point cloud method performs competitively with *Reproj. Equi*, underperforming slightly on the more challenging block stacking and block in bowl tasks. As observed in prior work (Wang et al., 2022b), the non-equivariant method underperforms equivariant methods even when the input group action is inaccurate (sideview).

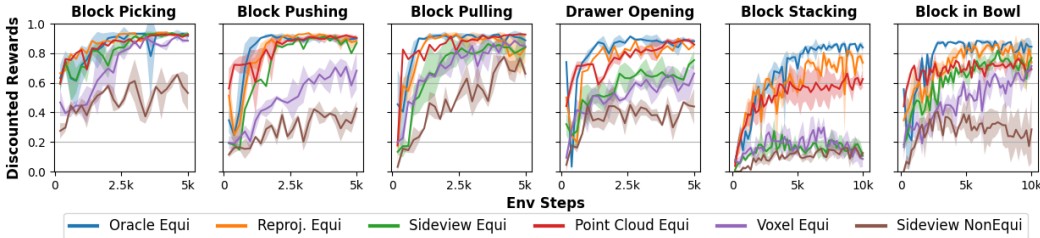

Figure 5: SACfD learning curves with RGBD images. Evaluation is performed every 200 environment steps. Results are averaged over three runs, shaded region shows standard deviation.

We observe the similar trends with behavior cloning. We compare performance on five environments with different amounts of demonstration data (5, 10 and 50 demos). The oracle observation achieves the highest success rate in nearly all settings as expected. Interestingly, using the reprojected observation slightly outperforms the oracle on block stacking (10 and 50 demos). Since block stacking requires accurately placing a block on top of another, the gripper holding the top block occludes the lower block in the oracle observation, which could make the policy harder to learn. This hypothesis is supported by the fact that the equivariant point cloud method performs best on this task since it operates directly on spatial data. When comparing across the number of demonstrations, we see that the re-rendering step outperforms the sideview most when there is less training data. With sufficient data, the equivariant network with sideview images learns to compensate for the inaccurate input group action, as discussed by Wang et al. (2024a).

Table 1: Success Rates (%) on Behavior Cloning trained on 5, 10 and 50 expert demonstrations. We report the maximum success rate achieved during training. Results are averaged over three seeds.

|  | Block Picking | | | Block Pushing | | | Drawer Opening | | | Block Stacking | | | Block in Bowl | | |
|---|---|---|---|---|---|---|---|---|---|---|---|---|---|---|---|
|  | 5 | 10 | 50 | 5 | 10 | 50 | 5 | 10 | 50 | 5 | 10 | 50 | 5 | 10 | 50 |
| Oracle Equi | 75.3 | 77.8 | 92.7 | 64.6 | 76.9 | 89.4 | 79.1 | 88.9 | 92.9 | 16.9 | 36.5 | 54.5 | 70.9 | 70.2 | 75.7 |
| Reproj. Equi | 77.0 | 84.7 | **92.5** | **75.4** | **86.5** | **92.0** | **70.1** | **80.7** | 91.9 | 24.5 | 41.6 | 56.5 | 61.8 | 66.1 | 74.4 |
| Sideview Equi | 49.9 | 75.2 | 92.1 | 42.5 | 59.1 | 89.5 | 46.2 | 66.6 | 89.2 | 6.3 | 16.9 | 30.2 | 29.6 | 50.0 | 68.6 |
| Voxel Equi | 65.6 | **85.2** | 91.2 | 44.5 | 50.1 | 84.5 | 63.2 | 75.6 | 90.3 | 25.5 | 33.7 | - | 44.0 | 55.6 | 68.1 |
| Point Cloud Equi | **79.3** | 83.3 | 85.8 | 73.8 | 82.7 | 91.5 | 51.8 | 70.3 | **92.7** | **45.3** | **55.5** | **64.2** | **66.3** | **69.1** | **75.8** |
| Sideview NonEqui | 40.3 | 56.2 | 76.4 | 30.6 | 50.0 | 82.0 | 34.9 | 57.1 | 80.4 | 3.7 | 4.5 | 7.6 | 12.4 | 22.0 | 45.0 |

## 5.5 Evaluation with RGB Images

We will now discuss the experiments with RGB images, where our proposed preprocessing step is a perspective transformation. In comparison to the occlusions associated with reprojection, we expect that the distortions from perspective transformation could confuse the network.

The results for SACfD with RGB images is shown in Figure 6. Using perspective transformation increases the performance of the equivariant model on all tasks. The gap is smallest on the drawer opening task, which makes sense because the drawer is tall and ends up distorted by the perspective transform. The performance on behavior cloning (Table 2) further shows the benefits of the perspective transform. In two cases (blocking picking 10 demos and block stacking 50 demos), the equivariant network with perspective transform even outperforms the oracle. However, for block pushing and drawer opening, the sideview image outperforms the perspective transformed image when trained on only five or ten demonstrations.

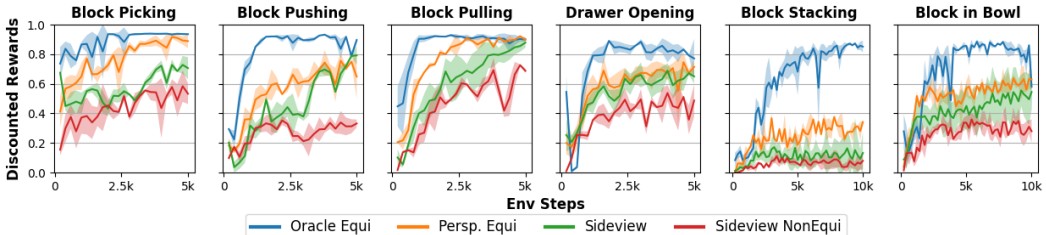

Figure 6: SACfD learning curves with RGB images. Evaluation is performed every 200 environment steps. Results are averaged over three runs, shaded region shows standard deviation.

Table 2: Success Rates (%) of behavior cloning with RGB images, given 5, 10 and 50 demonstrations. We report the maximum success rate achieved during training, averaged over three seeds.

| | Block Picking | | | Block Pushing | | | Drawer Opening | | | Block Stacking | | | Block in Bowl | | |
|---|---|---|---|---|---|---|---|---|---|---|---|---|---|---|---|
| | 5 | 10 | 50 | 5 | 10 | 50 | 5 | 10 | 50 | 5 | 10 | 50 | 5 | 10 | 50 |
| Oracle Equi | 80.9 | 76.0 | 93.8 | 61.1 | 79.8 | 91.0 | 79.4 | 87.8 | 92.7 | 19.8 | 29.9 | 30.0 | 64.9 | 67.4 | 75.7 |
| Persp. Equi | **66.4** | **85.2** | **93.0** | **72.4** | **82.3** | **90.6** | **54.5** | **71.5** | **89.9** | **13.5** | **30.3** | **56.7** | **49.8** | **57.5** | **73.7** |
| Sideview Equi | 52.5 | 72.0 | 89.1 | 46.2 | 57.6 | 82.5 | 37.5 | 56.6 | 85.7 | 6.1 | 10.8 | 20.6 | 18.0 | 35.9 | 56.0 |
| Sideview NonEqui | 35.6 | 60.2 | 72.0 | 22.5 | 43.1 | 69.6 | 34.9 | 47.0 | 78.0 | 3.3 | 4.1 | 5.0 | 10.5 | 16.3 | 36.5 |

## 5.6 EFFECTS OF CAMERA ANGLE

In this section, we demonstrate that the proposed preprocessing steps are effective over a range of camera view angles. The preprocessing steps are general, so they can be applied in any setting where the camera parameters (intrinsic and extrinsic) are known. However, the farther the camera moves from a top-down view, the more the processed image will contain occluded regions or distortions.

To understand how sensitive the method is to camera viewpoint, we recreate the behavior cloning experiments from earlier, but modulate the viewing angle of the camera from 15 to 75 degrees above horizontal. The results are reported in Figure 7 for both RGBD and RGB images, using ten demonstrations. We find that the equivariant network achieves higher success rates as the view angle increases. This was previously observed by Wang et al. (2022b) with equivariant SAC. Since the equivariant networks we use assume that the problem symmetry is described as rotations of the image, it makes sense that performance increases as the image appears more top-down.

With our preprocessing steps, the equivariant networks achieve strong performance at lower camera view angles. At a view angle of 75 degrees, the robot arm often occludes large portions of the image, which is why the non oracle methods do not appear to converge to the oracle at 90 degrees as one would expect. On block stacking, the equivariant method with our preprocessing outperforms the oracle. We believe this is because the sideview perspective is more informative to solve this task (with a top-down view the top block occludes the lower block during placement).

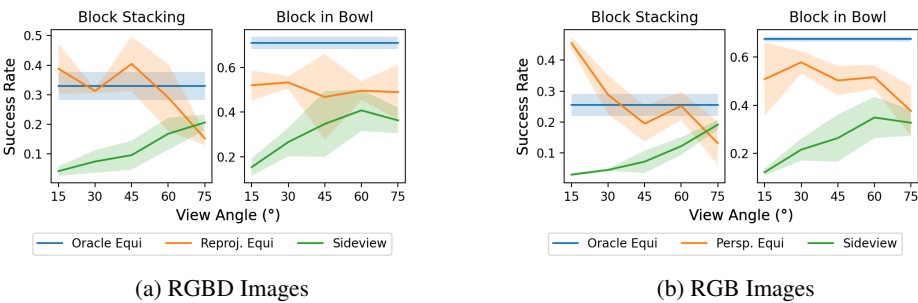

(a) RGBD Images              (b) RGB Images

Figure 7: Effect of View Angle on BC Performance. View angle is with respect to horizon (e.g. 90° is top-down view). Curves show average over three runs, shaded region is standard deviation.

## 5.7 Breaking or Relaxing Network Equivariance

The reprojection and perspective transform operations discussed in Section 4 introduce artifacts into the image, either due to missing pixels or distortion. These artifacts are linked to the original camera view since occlusions and distortions always occur on the far side of objects. As the images are processed with equivariant layers, the network will form representations of the rotated occlusions and distortions even though they would never rotate in real life (the front of an object is never occluded).

We proposed using interpolation to fill missing pixels and injecting view-direction information to enable the equivariant networks to learn effectively in the RGBD and RGB settings, respectively. Here, we empirically justify these steps by comparing versions of our method using networks with normal, broken and relaxed equivariance. For the relaxed version, we modify the equivariant network to use relaxed group convolutions introduced by Wang et al. (2022e). Relaxed group convolutions contain parameters which allow the model to deviate from perfect equivariance during training. The broken equivariance version is what we propose using with the perspective transform, where we add a standard representation ($pho_1$ feature) to the input that encodes the camera view direction. Since this additional input vector provides a fixed frame of reference, it breaks the symmetry of the equivariant model with respect to the observation. Additional details on these methods are in A.4.

Table 3: Effects of relaxing or breaking symmetry. Results are the maximum success rate (%) of a behavior cloning agent given ten demonstrations, averaged over three runs.

|      |                         | Drawer Opening | Block Stacking | Block in Bowl |
|------|-------------------------|----------------|----------------|---------------|
| RGBD | Reproj. Equi            | **80.7**       | **41.6**       | **66.1**      |
|      | Reproj. Equi (relaxed)  | 72.6           | 12.2           | 34.3          |
|      | Reproj. Equi (broken)   | 78.0           | 38.7           | 65.3          |
| RGB  | Persp. Equi.            | 71.5           | 30.3           | 57.5          |
|      | Persp. Equi (relaxed)   | 63.1           | 17.6           | 33.5          |
|      | Persp. Equi (broken)    | **76.0**       | **43.7**       | **65.5**      |

We find that relaxing or breaking equivariance in the RGBD setting results in worse performance of the policy (see Table 3). This suggests that the interpolation scheme effectively reduces the influence of the occluded regions on the learning. In contrast, we find that breaking equivariance results in the best performance in the RGB setting. In both settings, we find that the relaxed equivariance approach does the worst, despite significant effort to tune the hyperparameters. We believe relaxation could work in theory, but would have to be designed to specifically relax equivariance along a single direction to work effectively on this problem, where there is very little data with which to learn the relaxation weights.

## 6 Conclusion

In this work, we propose two simple preprocessing steps for $O(2)$ equivariant robot learning with sideview camera images. Since equivariant networks are designed for the case where the symmetry action is well-defined on the input space, the preprocessing steps transform the sideview images to look like images captured by a top-down camera. The first preprocessing step is reprojection, which can be applied to an RGBD image. Reprojection generates a partial point cloud from the image, then renders a new image with a virtual top-down camera frame. When depth information is not available, we propose using perspective transform instead, which maps visual information from the environment ground plane to a top-down camera frame.

**Limitations** We assume access to the camera extrinsics to perform the reprojection and perspective transformation. It may be difficult to estimate the camera pose with respect to the workspace when performing mobile manipulation or interacting with a complex workspace (e.g. no obvious ground plane). It may be possible to use a trainable reprojection operation, such as (Jaderberg et al., 2015; Gao et al., 2020), but it is unclear how well that would work with limited data. Another limitation is that the preprocessing steps do not fully close the performance gap compared to the top-down image case.

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

# A    APPENDIX / SUPPLEMENTAL MATERIAL

## A.1   EXAMPLE IMAGES BEFORE AND AFTER PREPROCESSING

In this section, we show example images of the manipulation environments before and after preprocessing. It may be easier to understand how the preprocessing aligns the image by seeing examples. Also, the visual artifacts from occlusion and distortion can be seen clearly (Figure 9 & 10).

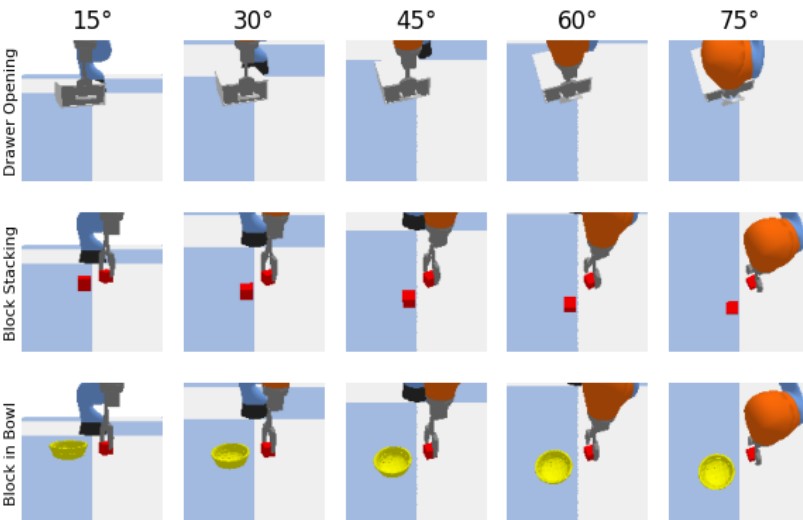

Figure 8: Example RGB images from sideview camera at different camera angles. Images taken from the middle of expert demonstration.

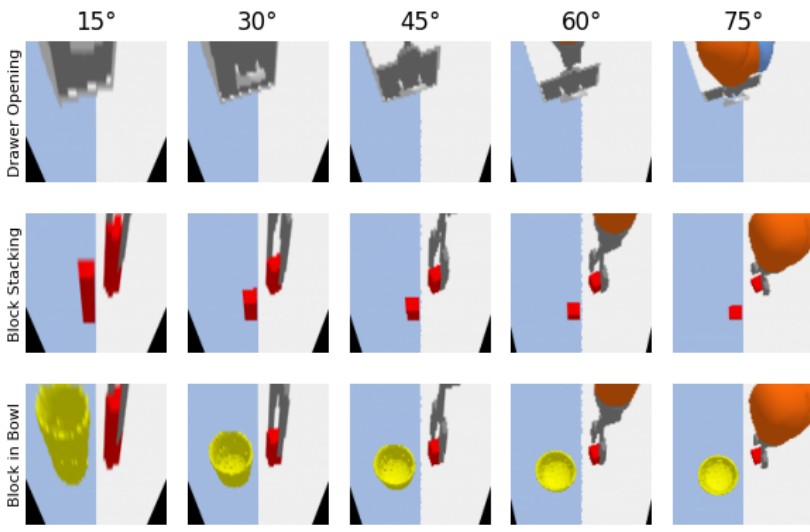

Figure 9: Example RGB images after perspective transform. The view angle of the sideview camera indicated at the top. Images taken from the middle of expert demonstration.

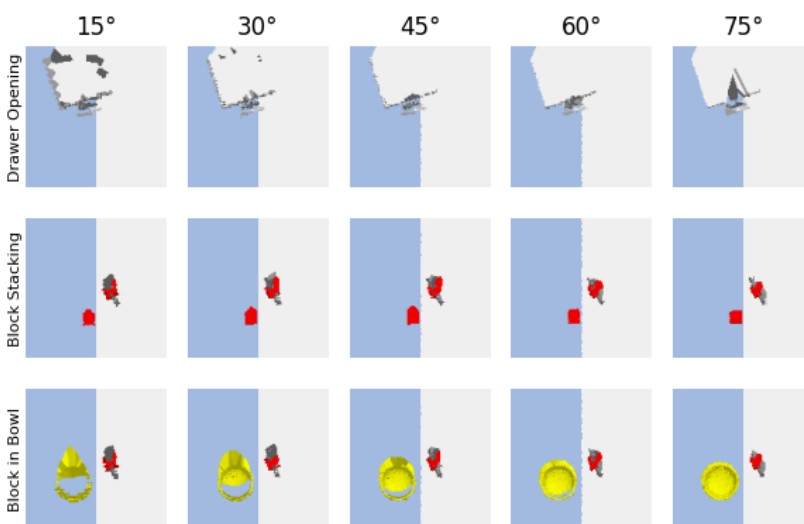

Figure 10: Example RGBD images after projection (depth channel not visualized here). The view angle of the sideview camera is indicated at the top. Images taken from the middle of expert demonstration. The occluded regions are inpainted with nearest neighbors (these regions are clearly visible with the yellow bowl).

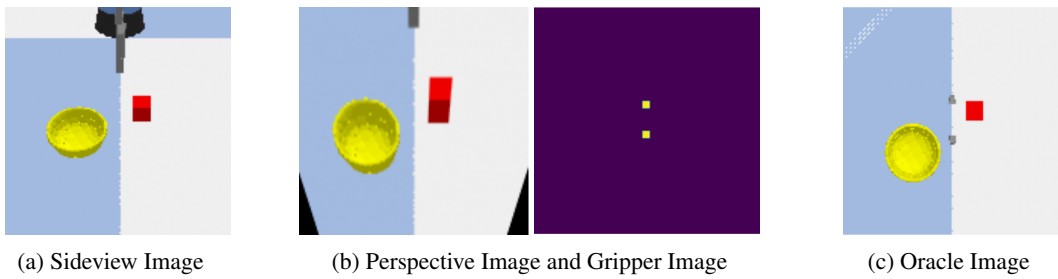

(a) Sideview Image                (b) Perspective Image and Gripper Image                (c) Oracle Image

Figure 11: Illustration of the gripper finger image that is added after performing a perspective transformation. When the sideview image is transformed, information about the position of the gripper is distorted, so we add an additional channel describing where the gripper fingers are in the top-down view.

## A.2 Environment Details

All manipulation environments are available in the BulletArm repository (Wang et al., 2022a). We use a perspective camera to render 152 by 152 images. The camera is positioned to capture the robot workspace with some padding on all sides, since we apply random crop augmentations (producing 128 by 128 images) during training. We add a channel to all images that contains the current gripper aperture.

## A.3 Training Details

**Soft Actor Critic from Demonstrations (SACfD)**   We follow the implementation of SACfD from (Wang et al., 2022c). We use a learning rate of 0.001, a batch size of 128, a discounting factor of 0.99, a target update interval of 100 optimization steps, a weighting of 1.0 on the expert action loss, and the Adam optimizer with default parameters. We use separate actor and critic networks. The number of demonstrations and training steps varies by environment (this is common practice from earlier work such that there is convergence across tasks of different difficulty). We use 20 demonstration episodes and 5,000 environment steps for all tasks except block in bowl and block

stacking, which are given 50 demonstrations and 10,000 environment steps. We run five simulators in parallel during training, and perform a single optimization step after every joint environment interaction (5 to 1 ratio of environment steps to optimization steps). We use a prioritized experience replay buffer (Schaul et al., 2015) to store the transitions with parameters: $\epsilon = 1e - 6$, $\alpha = 0.6$ and $\beta = 0.4$. We randomly rotate transitions four times when adding to the replay buffer, and apply another random crop augmentation when sampling from the buffer similar to RAD (Laskin et al., 2020). We evaluate the learned policies every 200 environment steps throughout training by averaging the discounted sum of rewards over 100 episodes (this is what we show in the learning curve plots).

**Behavior Cloning (BC)**    The policy network in behavior cloning is optimized with a mean squared loss between the expert action and the predicted action. We use a learning rate of 0.001, a batchsize of 128, and the Adam optimizer with default parameters. We train for 5,000 optimization steps for all BC experiments. Every 200 optimization steps, we evaluate the learned policy by averaging the success rate over 100 rollouts. In the tables, we report the maximum success rate achieved throughout training.

Our experiments were run on NVIDIA RTX 2080 GPUs with 12Gb of GPU RAM. Each run takes around 2-4 hours on a single GPU, depending on whether it is 5K or 10K steps.

A.4  NETWORK DETAILS

The SACfD agent has an actor and a critic network. The behavior cloning agent uses the same actor network architecture as SACfD. The actor and critic networks share similar architectures: an image encoder followed by an MLP head.

**Equivariant Image Encoder** The equivariant encoder is a $D_4$-equivariant convolutional network implemented with escnn (Cesa et al., 2022; Weiler & Cesa, 2019). The encoder has seven 2D convolution layers, with ReLU activations, and maxpooling such that the 128-by-128 input image is gradually downsampled to a 1x1 feature map. The input representation is trivial, and all internal representations are regular. The output of the network is a 128-dimensional regular representation. The equivariant image encoder has 1.1 M trainable parameters.

**Equivariant Actor** The equivariant actor is composed of an equivariant image encoder (described above) and a final equivariant linear layer. This final layer is $D_4$-equivariant that maps from a 128-dimensional regular representation to the action representation: $(a_x y, a_z, a_\theta, a_\lambda)$, where $a_x y$ is a $\rho_1$ representation (2D vector) and $a_z, a_\theta, a_\lambda$ are $\rho_0$ representations (scalars). This is the actor network used by *Sideview Equi*, *Oracle Equi*, *Reproj. Equi*, *Persp. Equi*, for all experiments.

**Equivariant Critic** The equivariant critic includes the equivariant image encoder and two MLP heads. The action is concatenated to the output of the image encoder (this forms a mixed representation: 128 regular representations and the action representation described above). This mixed representation is processed independently by two MLPs. Each MLP is a $D_4$ equivariant linear layer, followed by a ReLU, a group pooling operation, and a final $D_4$ equivariant layer. The output of each MLP is a scalar value describing the value of the state-action pair. SAC uses the minimum of the two predicted state-action values to update the critic and actor parameters (Haarnoja et al., 2018). This is the critic network used by *Sideview Equi*, *Oracle Equi*, *Reproj. Equi*, *Persp. Equi*, for all experiments.

**Non Equivariant Image Encoder** The non equivariant image encoder shares the exact same architecture as the equivariant encoder, where all convolutional layers are standard 2D convolutions. The size of the hidden dimensions inside the network are adjusted such that number of trainable parameters matches the equivariant encoder (1.1 M parameters).

**Non Equivariant Actor** The non equivariant actor is composed of the non equivariant image encoder and a linear layer. The linear layer predicts the five-dimensional action (plus an additional five values for the standard deviation when using SACfD). This is the actor network used by *Sideview NonEqui* in all experiments.

**Non Equivariant Critic** The non equivariant critic is composed of the non equivariant image encoder and two MLPs. The action is concatenated to the output of the image encoder and then pro-

cessed separately by two MLPs. Each MLP has two linear layers separated by a ReLU activation. This is the critic network used by *Sideview NonEqui* in all experiments.

**Equivariant Point Cloud Encoder** The architecture evaluated in Table 1 is described here. We implement a $D_4$-equivariant PointNet++ architecture (Qi et al., 2017), where all linear layers in the original network are replaced with equivariant versions. We use regular representations throughout the network, so equivariance is preserved through the ReLU activations and pooling schemes. The network is composed of three point set abstraction blocks with multi-scale grouping, and one point set abstraction block with single scale grouping, and a final linear layer that produces the equivariant action representation described in the equivariant actor above. The dimensionality of the hidden dimensions is modified so the number of trainable parameters is comparable to the other baselines (1.2 M parameters). The point cloud network runs significantly slower than the convolutional architectures, and a batchsize of 128 does not fit in 12Gb of GPU memory. We downsample the scene point clouds to 1024 points, and normalize the coordinates during training and evaluation.

**Equivariant Voxel Encoder** This network was constructed by modifying the architecture of the equivariant image encoder to use 3D convolutional layers instead of 2D convolutional layers. The 3D convolutional layers were instantiated using $D_4$ equivariance constraints on the weights. We reduced the number of channels at each layer such that the resulting encoder had comparable number of parameters to other baselines. Additionally, we set the strides and pooling in the z-direction such that the downsampling occurred gradually in all three dimensions. We generated voxel grids from RGBD observations using a grid size of 0.625cm, resulting in a 64x64x32 grid observation.

**Relaxed Equivariance** We use the same equivariant actor architecture but substitute all equivariant convolution layers in the encoder with relaxed group convolution layers (Wang et al., 2022e). We use a single filter bank to keep the number of parameters similar (the relaxed equivariant network ends up with 1.6 M parameters since it does not use steerable kernels). We tested a range of values for the regularization values: $\{1, 0.1, 0.01, 1e-4, 1e-6\}$ and $1e-4$ performed best.

**Broken Equivariance** We use the same equivariant actor network as above. The only modification is that we modify the input of the network to accept two additional channels. These channels contain a 2D vector pointing toward the camera viewpoint, and the network treats them as $\rho_1$ equivariant features. The idea is illustrated in Figure 12

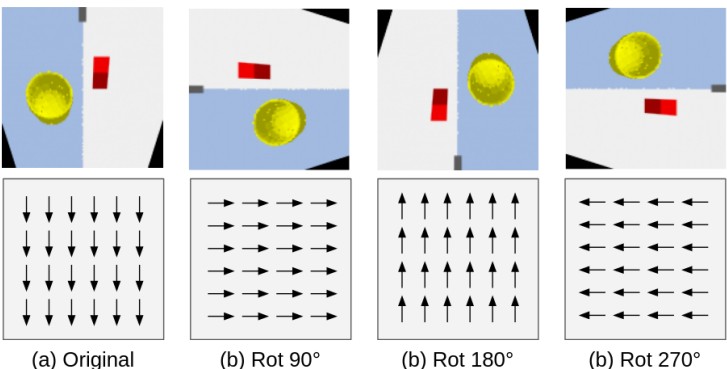

(a) Original      (b) Rot 90°      (b) Rot 180°      (b) Rot 270°

Figure 12: Illustrating symmetry breaking information, in the form of a 2D vector pointing toward camera. Any rotations of the image will also rotate the 2D vector, which means the equivariant netowrk can selectively process information based on the camera viewpoint (e.g. the distortion is always in the opposite direction of the camera viewpoint).

All networks were implemented and trained with PyTorch (Paszke et al., 2017).

## A.5 INTERPOLATION SCHEMES FOR RGBD REPROJECTION

In our RGBD reprojection approach, we infill occluded regions of the rendered image with interpolation. We conduct an experiment to understand the benefits of interpolating the occluded regions. The experiment compares nearest neighbor interpolation to local interpolation and no interpolation

on several behavior cloning tasks (see Table 4). We visualize each form of interpolation in Figure 13. Interpolations are performed independently on each channel of the RGBD image. The results show that using nearest neighbor interpolation achieves around 10% higher success rate than using local or no interpolation. We hypothesize that the occlusions in the reprojected image present a challenge to the rotationally-equivariant network that we use. Even though occlusions are directionally-biased (they are always on the side of the object away from the camera), the network must reason about them in all directions because of the equivariance constraints. For more complex scenes, nearest neighbor interpolation may be insufficient. In the future, we will explore using image inpainting models, which are very effective on realistic RGB images.

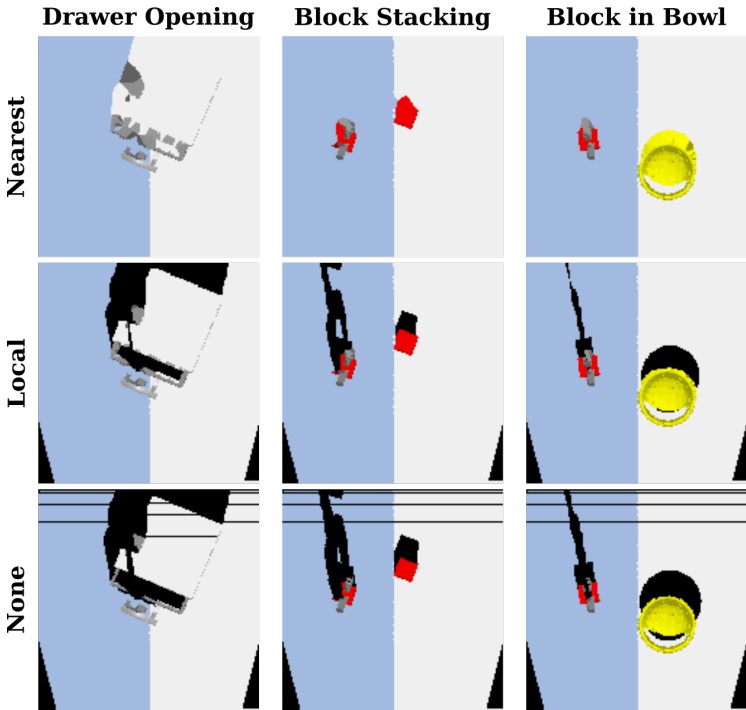

Figure 13: Illustration of interpolation schemes for reprojected image observations. Images are captured at a view angle of 45 degrees and then reprojected to a top-down view.

Table 4: Comparing Interpolation Schemes with RGBD Reprojection on BC tasks with ten demos. Results are reported as average best success rate over three runs.

|  | Drawer Opening | | | Block Stacking | | | Block in Bowl | | |
|---|---|---|---|---|---|---|---|---|---|
|  | 5 | 10 | 50 | 5 | 10 | 50 | 5 | 10 | 50 |
| Nearest Interpolation | 70.1 | 77.5 | 91.9 | 24.5 | 36.2 | 56.5 | 61.8 | 58.5 | 74.4 |
| Local interpolation | 44.5 | 69.4 | 91.2 | 6.0 | 31.3 | 61.7 | 44.1 | 54.1 | 69.9 |
| No interpolation | 52.0 | 69.3 | 89.8 | 11.9 | 25.2 | 42.4 | 41.0 | 49.2 | 71.8 |

## A.6 PERSPECTIVE TRANSFORM GROUND HEIGHT

The perspective transform assumes that all the image features live on a single plane, which we set to be the lower plane of the workspace, e.g. table surface. As we have already mentioned, this assumption is clearly violated for robotics tasks that contain 3D objects on top of the plane. Without access to depth information, we cannot identify and correct for distortions of the out-of-plane image features. Nevertheless, it is possible that the choice of ground plane may impact performance. We ran experiments to understand how the height of the plane used in the perspective transform affects performance. The results are reported in tab:sideview:persp-heights.

We find that the performance is robust to minor changes in the plane height of the perspective transform. For block stacking, the best performance is achieved with a plane height of 2.5 cm, which is roughly the height of the block. For drawer opening and block in bowl, the performance is stable up to 5.0 cm above the table surface. We believe any plane height that is within the working area of the task, e.g. the space where the dexterous behavior occurs, should work well. We also evaluated a relative plane height, where the plane height is dynamically set to be the height of the gripper fingers. A relative plane height does not work well at all, likely because the way objects distort changes based on robot actions, which confuses the convolutional network. We hope to investigate this setting more deeply in the future, since it could be useful for robotic manipulation tasks where there is no obvious working surface, such as handing off objects between manipulators.

Table 5: Comparing Ground Plane Height in Perspective Transform. BC with 10 demos

| Persp. Plane Height | Drawer Opening | Block Stacking | Block in Bowl |
|---|---|---|---|
| 0.0 cm | 76.0 | 43.7 | 65.5 |
| 2.5 cm | 76.4 | 47.5 | 65.0 |
| 5.0 cm | 76.5 | 35.6 | 63.3 |
| dynamic | 11.2 | 0.0 | 0.5 |

