# OpenReview forum: "Reducing Symmetry Mismatch Caused by Freely Placed Cameras in Robotic Learning"
_ICLR.cc/2025/Conference — Submitted to ICLR 2025_

### Official Review · Reviewer_FKHM · 2024-10-20

**Soundness:** 3
**Presentation:** 3
**Contribution:** 1
**Rating:** 3
**Confidence:** 4

**Summary:**

The paper addresses a challenge in robotic learning, where freely placed cameras cause a mismatch between input image transformations and the inherent task symmetry in robotic manipulation environments. The authors propose two preprocessing methods: reprojection of RGBD images and perspective transformation for RGB images. These techniques transform side-view images into top-down views, thus aligning the image transformations with the task symmetry. This approach is shown to consistently improve performance in robotic manipulation tasks, particularly in reinforcement learning and imitation learning setups.

**Strengths:**

1. The paper offers practical preprocessing methods (RGBD reprojection and RGB perspective transformation) that are simple. These methods can be applied across various robotic learning tasks without additional training or modification of the robot setup.

2. The proposed methods require only knowledge of the camera’s intrinsics and extrinsics, making them straightforward to implement without the need for privileged information. This makes the approach broadly applicable across robotic tasks.

**Weaknesses:**

1. Limited Technical Contribution: The technical contribution of the paper is minimal. The methods of RGBD reprojection and RGB perspective transformation are well-established and mature techniques. The paper merely applies these existing methods to Equivariant Policy Learning without introducing any significant novel ideas. As a result, the work feels more like a technical report rather than a research paper offering new scientific insights.

2. Lack of Real-World Experiments: The experiments are conducted only in six simple simulated environments, without any real-world validation. This limits the applicability and robustness of the proposed methods in practical scenarios, as real-world experiments are essential to demonstrate the effectiveness of the approach outside of controlled simulations.

3. Performance Gap with Oracle: While the proposed methods reduce the performance gap with the oracle top-down view, they do not entirely close it. The occlusion of objects and grippers, especially in cluttered environments, remains an unsolved problem.

**Questions:**

1. Handling Extreme Occlusions: In the RGBD setting, how might more sophisticated inpainting or occlusion handling methods (e.g., learned inpainting) improve the performance gap with the oracle? Have the authors experimented with these techniques, and what were the results?

2. Effectiveness in Real-World Scenarios: While the experiments are simulated, can the authors elaborate on the challenges and potential modifications required to apply these preprocessing steps in real-world robot learning tasks with physical cameras and hardware?

---

> ### Author Response · Authors · 2024-11-25
>
> We thank the reviewer for their helpful comments.
>
> > The experiments are conducted only in six simple simulated environments, without any real-world validation.
>
> We agree that real-world validation would strengthen this method.  We are not able to provide results for real world tasks during the rebuttal period.
>
> > While the proposed methods reduce the performance gap with the oracle top-down view, they do not entirely close it. The occlusion of objects and grippers, especially in cluttered environments, remains an unsolved problem.
>
> This is a good point.  Keep in mind that the oracle view is an unrealistic ideal setting, since its very challenging to achieve the oracle view in a real-world setting.  We are interested in closing the gap with the oracle further and are open to suggestions for experiments that would shed light on why this gap remains.
>
> > In the RGBD setting, how might more sophisticated inpainting or occlusion handling methods (e.g., learned inpainting) improve the performance gap with the oracle? Have the authors experimented with these techniques, and what were the results?
>
> There are many effective inpainting methods available that could be used in place of interpolation.  In our simulated results, interpolation worked well, due to the simplicity of the scene.  We believe these inpainting methods would be more useful in real-world tasks with more complex backgrounds and objects.
>
> > While the experiments are simulated, can the authors elaborate on the challenges and potential modifications required to apply these preprocessing steps in real-world robot learning tasks with physical cameras and hardware?
>
> We have begun real-world experiments in the RGBD setting.  The main difference we observed was noisiness in the depth values on the gripper.  After reprojection, the appearance of the gripper was very distorted (missing regions or missing entire finger), which we believe made it difficult for the model to learn.  One option to help with this is to add a synthetically rendered gripper image to the input (similar to how we do it for the RGB setting).

---

> > ### Comment · Reviewer_FKHM · 2024-11-27
> >
> > Thank you for your response, but I believe your reply did not address my main concern: the technical contribution of the paper is minimal. Therefore, I will maintain the score I gave.

---

### Official Review · Reviewer_aBnp · 2024-11-03

**Soundness:** 3
**Presentation:** 3
**Contribution:** 1
**Rating:** 3
**Confidence:** 4

**Summary:**

This paper addresses the limitations of a certain type of equivariant policy learning in robotic manipulation tasks when using side-view camera perspectives, which cause symmetry mismatches that reduce performance. The authors propose a simple method to transform side-view images into top-down representations, enhancing the performance of equivariant methods. Its effectiveness is demonstrated on RGB and RGBD images.

**Strengths:**

- The problem formulation is quite straightforward. The proposed method is simple, intuitive, and effective.
- The discussion on Occluded Regions for RGBD images and Out-of-plane Distortion for RGB images makes the proposed method more practical, giving it the potential to be deployed in real-world settings.

**Weaknesses:**

- Though very simple and effective under the tested scenario, this paper seems more like a small pre-processing module specifically designed for a certain type of SO(2) RL and IL methods. How many equivariant methods could benefit from the proposed method? I would like the authors to discuss this question, and list as many papers as possible.
- Lacking real-world experiments. I am concerned whether the proposed method would be effective as well in real-world settings. And since the proposed method is mainly designed to tackle the challenge when deploying cameras in the real world, I think real-world experiments are indispensable.

**Questions:**

- How many equivariant methods could the proposed method benefit?
- Would the proposed method also benefit general-purpose robot learning methods such as Diffusion Policy?
- It seems that the compared point cloud baseline is using a single-view RGBD image. What if we have access to multi-view RGBD images? Consider the scenario in [1].
- Would real-world experiments be conducted?


[1] RiEMann: Near Real-Time SE (3)-Equivariant Robot Manipulation without Point Cloud Segmentation. Gao et al. CoRL'24.

---

> ### Author Response · Authors · 2024-11-25
>
> We thank the reviewer for their helpful comments.
>
> > How many equivariant methods could benefit from the proposed method? I would like the authors to discuss this question, and list as many papers as possible.
>
> The point of this work is not to improve existing equivariant methods.  In fact, we believe existing equivariant methods work very well within structured settings (e.g. those where the symmetric transformation of the observation is well-defined).  Instead, we show that image-based equivariant methods can also excel in less structured settings (e.g. those with non top-down camera views) when they incorporate the simple pre-processing steps.  Here is a list of robotic manipulation papers that fit into this less structured category, where our work could be beneficial
>
> > since the proposed method is mainly designed to tackle the challenge when deploying cameras in the real world, I think real-world experiments are indispensable
>
> We agree that real-world results would strengthen the paper.  We are not able to provide these results within the discussion period.
>
> > Would the proposed method also benefit general-purpose robot learning methods such as Diffusion Policy?
>
> We believe the proposed method would help any SO(2) equivariant policy learning methods with non-top down images.  The processed pre-processing steps align transformations of the input with transformations of the action, which leads to faster learning.  As shown by a recent paper [1], SO(2) equivariant diffusion policy is useful for robotic manipulation, and we expect our preprocessing steps could enhance performance with sideview image inputs.
>
> > It seems that the compared point cloud baseline is using a single-view RGBD image. What if we have access to multi-view RGBD images? Consider the scenario in [1].
>
> That is a good question.  In theory, multiple RGBD images could be fused and reprojected to produce a top-down image.  The resulting top-down image would have less occluded regions than the single-view RGBD case, which would boost performance.  However, in the multi-view RGBD setting, the equivariant point cloud method is probably a better option since it can process a complete point cloud effective (reprojection collapses the information in the z-direction).   We are interested in looking more into the multi-view RGB setting, such as the ALOHA system.
>
> [1] Wang, Dian, et al. "Equivariant diffusion policy." arXiv preprint arXiv:2407.01812 (2024).

---

### Official Review · Reviewer_vGKo · 2024-11-03

**Soundness:** 3
**Presentation:** 2
**Contribution:** 2
**Rating:** 3
**Confidence:** 4

**Summary:**

The paper proposes a method to improve equivariant policy learning in robotic manipulation tasks where camera views are not ideal (e.g., side views instead of top-down). The authors present two preprocessing techniques:
- Reprojection of RGBD images to approximate top-down views by generating point clouds and interpolating missing data.
- Perspective transformation of RGB images to map the ground plane onto a top-down view.

These methods enhance performance across different learning tasks and camera angles without additional data or privileged information, making them adaptable to real-world setups. The experiments show improved policy learning outcomes in several robotic tasks by aligning image transformations with physical symmetries in the robot workspace.

**Strengths:**

- The paper defines a problem of "symmetry mismatch" from non-ideal camera placements in image based equivariant robotic learning. By applying reprojection and perspective transformations to side-view images, it extends the utility of equivariant learning in robotics, enabling its application in more realistic setups.
- The authors provide a thorough and well-validated empirical analysis across diverse robotic tasks and modalities (RGB and RGBD), with clear comparisons to multiple baselines. This experimental rigor strongly supports the paper's claims about the effectiveness of the preprocessing techniques.

**Weaknesses:**

- The problem this paper attempts to address may not be a genuine issue. When handling tabletop robotic manipulation tasks and aiming to apply O(2)-equivariant policy learning algorithms, a fundamental assumption is the availability of top-view observations. If only side-view images are accessible, a more natural approach might be to consider non-equivariant policy learning algorithms instead.

- The methods proposed in this paper lack originality. Both 3D reprojection and perspective transformation are well-established algorithms in the field of computer vision. This paper merely applies them to a specific scenario—converting side-view images of tabletop robotic manipulation scenes into top-view images—to facilitate the use of O(2)-equivariant policy networks. I view these techniques as pre-processing tricks rather than substantive innovations.

- The formulation for 3D reprojection in this paper is not entirely realistic. To perform reprojection, RGBD information is required. However, if 3D data is available, it would be more straightforward to use equivariant policy networks based on 3D groups$^{[1,2]}$ (such as SO(3), SE(3), or SIM(3)). This would eliminate the need to address issues arising from mismatched camera viewpoints.

[1] Yang, J., Cao, Z. A., Deng, C., Antonova, R., Song, S., & Bohg, J. (2024). Equibot: Sim (3)-equivariant diffusion policy for generalizable and data efficient learning. arXiv preprint arXiv:2407.01479.

[2] Chen, Y., Tie, C., Wu, R., & Dong, H. (2024). EqvAfford: SE (3) Equivariance for Point-Level Affordance Learning. arXiv preprint arXiv:2408.01953.

**Questions:**

- In the experimental part, the author compares many baselines (equivariant, non-equivariant, 2D, 3D), but does not clearly write out the specific structure of each baseline and the group on which their equivariance properties are defined.
- All baseline methods in this paper are based on the same framework (SACfD). To demonstrate the effectiveness of this preprocessing approach in broader scenarios, I believe it would be beneficial to include comparisons with other state-of-the-art methods.

---

> ### Author Response · Authors · 2024-11-25
>
> We thank the reviewer for their helpful comments.
>
> > The problem this paper attempts to address may not be a genuine issue… a more natural approach might be to consider non-equivariant policy learning algorithms instead.
>
> We directly compare our proposed approach against non-equivariant policy learning algorithms. The non-equivariant baselines perform much worse in terms of performance and sample efficiency (see “Sideview NonEqui” Figure 5 and Table 1).  The non-equivariant methods were trained with data augmentation and still underperformed the equivariant versions.  These results were also observed in [1].
>
> > The methods proposed in this paper lack originality… I view these techniques as pre-processing tricks rather than substantive innovations.
>
> We agree that these techniques are simple and well-known, and we tried to make that clear in the writing.  We see this simplicity as a benefit to our approach since it expands the problem settings where equivariant policy learning methods are useful without requiring different sensors or new network architectures.
>
> > The formulation for 3D reprojection in this paper is not entirely realistic. To perform reprojection, RGBD information is required.
>
> We believe the reviewer is misunderstanding our work.  We only perform 3D projection when RGBD information is available.  When it is not available, we perform a perspective transform, which, as we note in the paper, deviates from a 3D reprojection for any visual features that are above the ground plane.
>
> >if 3D data is available, it would be more straightforward to use equivariant policy networks based on 3D groups$^{[1,2]}$ (such as SO(3), SE(3), or SIM(3))
>
> You are correct.  If we use an SO(3) equivariant network, then the original point cloud (in the sideview camera frame) could be used directly as input.  In that setting, we would have to apply pooling at the end to reduce to an SO(2) equivariant representation for the output actions.  One downside to this approach is the additional compute.  Moving from SO(2) to SO(3)/SE(3) equivariance requires applying additional constraints that slow down training and increase memory.  The other downside is that an SO(3)/SE(3) equivariant network cannot resolve the “gravity” direction (they are invariant to the input’s coordinate frame).  In many robotic manipulation tasks, the “gravity” direction is important for determining the best action.  There are ways to re-inject information about gravity, but it would take some experimentation to identify the best approach.
>
> We believe running an experiment would be interesting.  We are training a VectorNeuron [1] baseline and will try to add the results by the end of the discussion period.
>
> > In the experimental part, the author compares many baselines (equivariant, non-equivariant, 2D, 3D), but does not clearly write out the specific structure of each baseline and the group on which their equivariance properties are defined.
>
> We include the symmetry group used and the specific structure of all networks in Appendix A.4.  We refer to this part of the Appendix at the end of Baselines (Section 5.3).
>
> > All baseline methods in this paper are based on the same framework (SACfD). To demonstrate the effectiveness of this preprocessing approach in broader scenarios, I believe it would be beneficial to include comparisons with other state-of-the-art methods.
>
> In this paper, we run experiments with SACfD (Figure 5 & 6) and behavior cloning (Table 1 & 2).  Do you have a SOTA method in mind that we should run comparisons on?
>
> [1] Deng, Congyue, et al. "Vector neurons: A general framework for so (3)-equivariant networks." Proceedings of the IEEE/CVF International Conference on Computer Vision. 2021.

---

> > ### Comment · Reviewer_vGKo · 2024-11-26
> >
> > I do not think the response of the authors effectively address my concerns.
> >
> > This article uses two well-known techniques in computer vision as preprocessing for the input of equivariant networks. I think this work lacks innovation and is just a preprocessing technique. At the same time, the author has not verified the applicability range of this technique. Almost all experiments are based on the same algorithm framework (SACfD), and this technique has not been applied in more equivariant learning methods that require image input to verify its effect. In addition, the author has not verified in real-world input whether their proposed technique can improve existing methods. Therefore, combined with the opinions of other reviewers, I think the quality of this article is not sufficient to be accepted by ICLR.

---

### Official Review · Reviewer_sTng · 2024-11-03

**Soundness:** 3
**Presentation:** 3
**Contribution:** 2
**Rating:** 5
**Confidence:** 4

**Summary:**

This paper addresses a key issue in equivariant neural networks for agent learning to decrease the gap between sideview camera observations, which perform sub-optimally when cameras view the scene from the side rather than directly above. The authors propose two simple preprocessing techniques to reduce this gap:

1. For RGBD cameras, they reproject the image to a virtual top-down view, and
2. For RGB cameras, they apply a perspective transformation to align the ground plane with the image plane.

Through experiments across multiple robotic manipulation tasks using both reinforcement learning and imitation learning, they demonstrate that these preprocessing steps significantly improve the performance of equivariant networks compared to using raw side-view images.

**Strengths:**

1. This work addresses a very common problem prevailing in the robotic manipulation domain i.e lack of robustness of vision based policies to viewpoints.
2. The proposed solutions are very simple (under the known camera extrinsics assumption, which is typically common in table-top robotic manipulation settings).
3. The paper is generally well written and easy to understand in a single go.

**Weaknesses:**

4. Related works: I believe a small discussion on point cloud models (in the context of Image reprojection) should also be discussed. Several works in the the past few years have proposed using point clouds for RL / policy learning [1, 2] and shown robustness to viewpoints [3].

5. Sample-efficiency of RGBD experiments: I don't particularly find a difference between *Point cloud equi* and *Reproj. equi* in Fig 5. and Table 1. What are the benefits of Reproj. Equi over point cloud equi?

6. Sec 5.6 (Effects of camera angle) needs to also have the PointNet++ baseline (*point cloud equi*) for the RGB-D plots. Some works have suggested that point cloud RL policies are robust to viewpoint changes [3].
---
**References:**

1. On the efficacy of 3d point cloud reinforcement learning, Zhan Ling et al., arXiv 2023
2. Point Cloud Matters: Rethinking the Impact of Different Observation Spaces on Robot Learning, Haoyi Zhu et al., NeurIPS D&B 2024.
3. Point Cloud Models Improve Visual Robustness in Robotic Learners, Skand Peri et al., ICRA 2024

---
**Rationale for current rating**: Overall I believe this is a well written paper with clear contributions. However, I have particular questions regarding the baselines (points 5, 6, 8) and generalization (point 9) and based on that, I'm voting for a weak reject. However, this is *not* my final decision and I am willing to update my score based on other reviewers' comments and authors' rebuttal.

**Questions:**

7. Gripper Image: Does this formulation of having a gripper image generalize to a dextrous manipulation with a non-trivial gripper? Also, it would be better if the Fig 11 (from appendix) can be moved/integrated into the main paper. This is because, the gripper representation is one of the crucial aspects of the proposed solution and having it in a visual form would make the methodology more clearer to the reader.

8. I would like to see an experiment with DrQ-v2 where image augmentaiton has shown significant sample-efficiency gains and am curious how that performs as compared to an explicit Equivariant policy. I believe the data-augmentations can be implemented in a straightforward manner within the SACfD codebase.

9. Are the models in Fig 7(a) and 7(b) test-only models are are they trained on individual camera angles? If it's trained and tested separately -- I'm curious to see how Reproj equi or Presp. equi perform on testing on OOD camera viewpoints (i.e train on one camera angle and test on rest.)

10. Are the class of Equivariant policies biased to the action space? Would the same set of architectures work for a other action spaces that are common in robotic manipulation such as end-effector pose, joint velocities, joint angle poisitions etc?

---

> ### Author Response · Authors · 2024-11-25
>
> We thank the reviewer for their helpful comments.
>
> > Related works: I believe a small discussion on point cloud models (in the context of Image reprojection) should also be discussed.
>
> We will add a discussion on point cloud models in the updated version of the paper.
>
> > I don't particularly find a difference between Point cloud equi and Reproj. equi in Fig 5. and Table 1. What are the benefits of Reproj. Equi over point cloud equi?
>
> The performance of Reproj Equi and Point Cloud Equi are comparable.  In general, point cloud networks are more compute and memory-intensive (more so when enforcing equivariance constraints), so we downsample the input to 1024 or 2048 points.  So in settings where the task requires high-resolution observations (like peg-insertion or grasping mug handle), the point cloud model may struggle.  In contrast, Reproj Equi uses an image encoder that can handle high-resolution inputs.
>
> > Sec 5.6 (Effects of camera angle) needs to also have the PointNet++ baseline (point cloud equi) for the RGB-D plots. Some works have suggested that point cloud RL policies are robust to viewpoint changes
>
> We agree that this would be a good comparison.  We will try to add the result by the end of the discussion period.
>
> > I would like to see an experiment with DrQ-v2 where image augmentaiton has shown significant sample-efficiency gains and am curious how that performs as compared to an explicit Equivariant policy.
>
> Image augmentation is helpful for equivariant and non-equivariant policy learning, as shown by [1].  In our paper, all methods apply random image crops to observations during training as is the case in DrQ-v2.  In the paper, we cite this technique as RAD [2], which was a simpler, concurrent work to DrQ (v1).
>
> >Does this formulation of having a gripper image generalize to a dextrous manipulation with a non-trivial gripper?
>
> This is a great question.  The gripper image is generated with the assumption that the gripper model is known.  So generating the gripper image is possible as long as the gripper is fully actuated and composed of rigid parts, regardless of how dexterous the task is.
>
> > it would be better if the Fig 11 (from appendix) can be moved/integrated into the main paper.
>
> We agree.  We have added the gripper image to Figure 3, which is next to where we introduce the gripper image.
>
> > Are the models in Fig 7(a) and 7(b) test-only models are are they trained on individual camera angles?
>
> The models in Figure 7 are tested on the same view angle as they were trained on.  It would be interesting to see if the models could generalize to novel view angles (since the projection and perspective transform approximately canonicalize the perceived view to be topdown).
>
> > Are the class of Equivariant policies biased to the action space? Would the same set of architectures work for a other action spaces that are common in robotic manipulation such as end-effector pose, joint velocities, joint angle poisitions etc?
>
> Equivariant policy networks are specific to an action space.  Consider the equivariance equation (Eqn 1), we need to know the action of the group on the output space when we create the equivariant network.  If we change the output space (action space), then we need to modify the equivariant constraints at the end of the network accordingly. This is easy for action spaces based on end effector pose or velocity, but difficult for joint space control.  For instance, if we apply a 2D transformation to the scene, it is not clear what transformation should be applied to the joint angles to produce a similar action.  This would be an interesting direction to pursue in the future since there is growing interest in policy learning in joint space (like in the ALOHA system).
>
> [1] Wang, Dian, et al. "The Surprising Effectiveness of Equivariant Models in Domains with Latent Symmetry." The Eleventh International Conference on Learning Representations.
>
> [2] Laskin, Misha, et al. "Reinforcement learning with augmented data." Advances in neural information processing systems 33 (2020): 19884-19895.

---

### Meta-Review · Area_Chair_eAd8 · 2024-12-21

**Metareview:**

The paper addresses the performance limitations of equivariant policy learning in robotic manipulation when using side-view camera perspectives by identifying a "symmetry mismatch" between side-view inputs and the symmetry assumptions of equivariant networks. To overcome this, the authors propose two simple preprocessing techniques: reprojecting RGBD images to a virtual top-down view using depth data and applying perspective transformations to RGB images to align the ground plane with the image plane. These methods transform side-view images into top-down representations, thereby enhancing the effectiveness of equivariant policy networks.

Strengths:

-- Tackles a common issue in robotic manipulation regarding viewpoint robustness, making equivariant learning applicable to more realistic camera setups.

-- The preprocessing techniques are straightforward, requiring only known camera extrinsics, facilitating easy integration into existing systems.

-- Demonstrates consistent performance enhancements across multiple tasks and image modalities, providing strong empirical support for the proposed methods.

Weaknesses:

-- Utilizes established computer vision techniques (3D reprojection and perspective transformation) without introducing new algorithms or theoretical advancements.

--  Lacks thorough comparisons with state-of-the-art point cloud-based equivariant methods, weakening claims of superiority and generalizability.

-- Relies solely on simulated environments, lacking validation in real-world settings which is crucial for practical applicability.

After carefully reading the paper, the reviews and rebuttal discussions, the AC finds despite effectively addressing a practical problem and demonstrating consistent empirical improvements, the paper lacks significant technical innovation, fails to provide comprehensive benchmarking against advanced point cloud-based methods, and does not include real-world experiments.  AC agrees with the reviewers on recommending to reject the paper.

**Additional Comments On Reviewer Discussion:**

See the weakness and comments above, there are still remaining concerns from most reviewers.

---

### Decision · Program_Chairs · 2025-01-22

Reject